# Extracellular Vesicles in Ovarian Cancer: From Chemoresistance Mediators to Therapeutic Vectors

**DOI:** 10.3390/biomedicines12081806

**Published:** 2024-08-09

**Authors:** Barathan Muttiah, Nur Dina Muhammad Fuad, Faizul Jaafar, Nur Atiqah Haizum Abdullah

**Affiliations:** 1Department of Tissue Engineering and Regenerative Medicine, Faculty of Medicine, Universiti Kebangsaan Malaysia, Cheras, Kuala Lumpur 56000, Malaysia; p138566@siswa.ukm.edu.my; 2Jeffrey Cheah School of Medicine and Health Sciences, Faculty of Medicine, Monash University, Bandar Sunway, Subang Jaya 47500, Malaysia; faizul.jaafar@monash.edu

**Keywords:** ovarian cancer (OC), extracellular vesicles (EV), nanocarriers, chemotherapy resistance, drug delivery

## Abstract

Ovarian cancer (OC) remains the deadliest gynecological malignancy, with alarming projections indicating a 42% increase in new cases and a 51% rise in mortality by 2040. This review explores the challenges in OC treatment, focusing on chemoresistance mechanisms and the potential of extracellular vesicles (EVs) as drug delivery agents. Despite advancements in treatment strategies, including cytoreductive surgery, platinum-based chemotherapy, and targeted therapies, the high recurrence rate underscores the need for innovative approaches. Key resistance mechanisms include drug efflux, apoptosis disruption, enhanced DNA repair, cancer stem cells, immune evasion, and the complex tumor microenvironment. Cancer-associated fibroblasts and extracellular vesicles play crucial roles in modulating the tumor microenvironment and facilitating chemoresistance. EVs, naturally occurring nanovesicles, emerge as promising drug carriers due to their low toxicity, high biocompatibility, and inherent targeting capabilities. They have shown potential in delivering chemotherapeutics like doxorubicin, cisplatin, and paclitaxel, as well as natural compounds such as curcumin and berry anthocyanidins, enhancing therapeutic efficacy while reducing systemic toxicity in OC models. However, challenges such as low production yields, heterogeneity, rapid clearance, and inefficient drug loading methods need to be addressed for clinical application. Ongoing research aims to optimize EV production, loading efficiency, and targeting, paving the way for novel and more effective therapeutic strategies in OC treatment. Overcoming these obstacles is crucial to unlocking the full potential of EV-based therapies and improving outcomes for OC patients.

## 1. Introduction

Ovarian cancer (OC) stands out as the deadliest gynecological malignancy and the fifth most common cause of cancer-related deaths in women [1]. The global rise of OC poses a major public health threat. Projections estimate a staggering 42% increase in new cases by 2040, with diagnoses climbing from 313,959 in 2020 to a projected 445,721. This alarming trend extends to mortality rates as well, with a projected 51% rise in OC deaths, from 207,252 in 2020 to 313,617 by 2040 [2]. The most substantial absolute increase in cases is anticipated in Asia, reflecting the continent’s growing population. However, the most pronounced percentage increase is expected in Africa, where the number of women diagnosed with OC is projected to nearly double over the next two decades [3]. The projected increase in Asia could be driven by population growth, aging, and changing lifestyles, including diet, physical activity, and exposure to environmental risks [4]. In Africa, the doubling of cases could be influenced by improved diagnostic capabilities, demographic changes, and possibly increasing risk factors like urbanization and the westernization of diets [5]. The late diagnosis of OC contributes to its high mortality rate, with most cases detected at an advanced stage, leading to poor outcomes [6]. Another major challenge in addressing OC is its typically asymptomatic development until it has advanced, often characterized by ascites or metastasis [6,7]. This contributes to the difficulty in early diagnosis and results in a 5-year survival rate of less than 30% [6]. Existing early detection methods, like transvaginal ultrasound and CA-125 assays, have low predictive value, underscoring the challenges in diagnosing this disease early [8].

### 1.1. Current Treatment for OC

OC treatment has become increasingly diverse and advanced, with various strategies focused on enhancing clinical outcomes. Ovarian cancer treatment often starts with cytoreductive surgery, which prioritizes the removal of as much tumor tissue as feasible [9]. This can involve a range of procedures, such as hysterectomy, salpingo-oophorectomy, omentectomy, and lymph node biopsy, depending on the stage and spread of the cancer [10]. After surgery, platinum-based chemotherapy such as cisplatin and carboplatin are generally administered to eliminate any remaining cancer cells [11]. However, despite these intensive efforts, approximately 70% of OC patients experience a recurrence within 18 to 28 months, requiring additional treatment for recurrent ovarian cancer (ROC) [12].

For platinum-sensitive ROC, combination chemotherapy with agents like paclitaxel or gemcitabine in addition to carboplatin is common, whereas platinum-resistant ROC might require other agents such as topotecan [13]. In advanced OC or early-stage OC with high-grade histology, adjuvant chemotherapy is typically recommended. Beyond traditional chemotherapy, antiangiogenic agents like bevacizumab, which inhibit tumor angiogenesis through vascular endothelial growth factor (VEGF) blockade, have shown benefits in prolonging progression-free survival (PFS) in advanced OC [14].

PARP inhibitors, such as Olaparib and Niraparib, represent a breakthrough in OC treatment. These targeted agents have shown remarkable success in extending PFS, particularly for patients with breast cancer gene (BRCA) mutations. This development marks a significant shift towards personalized medicine, focusing on targeted maintenance therapy [15]. Immune checkpoint inhibitors (ICIs), including cytotoxic T-lymphocyte-associated protein 4 (CTLA-4) and programmed cell death protein 1 (PD-1) inhibitors, are also gaining traction for their potential to enhance the immune system’s response against cancer [16]. However, ICIs have shown limited efficacy in OC compared to other cancers like non-small-cell lung cancer and melanoma, with monotherapy response rates ranging from 10 to 15% [17,18]. Ongoing clinical trials are exploring the effectiveness of ICIs in combination with other treatments to improve outcomes for OC patients.

OC’s hormone dependency, which involves steroid hormones like estrogen and progesterone, suggests that hormone therapy could be a viable treatment strategy [19]. Aromatase inhibitors and anti-estrogen treatments like tamoxifen have shown some efficacy in hormone-dependent OC. Still, the complex interplay of hormonal pathways and their impact on OC progression requires further research [20]. The overall goal of OC treatment is to improve survival rates and enhance the quality of life for patients. However, the high rate of recurrence and treatment resistance observed in relapsed cases underscores the need for new approaches to prevention and treatment [21]. This situation highlights the urgency for global efforts to advance prevention, early detection, and innovative treatment strategies, providing hope for improved outcomes in the future. Figure 1 outlines the diverse and advanced treatment strategies for OC, emphasizing the various approaches taken to enhance clinical outcomes and improve patient survival rates.

### 1.2. Chemotherapeutic Resistance in OC

The emergence of drug resistance is a significant challenge in cancer therapy, particularly in the treatment of OC. Despite the initial efficacy of chemotherapeutic agents like cisplatin and paclitaxel, many patients eventually develop resistance, leading to treatment failure and tumor recurrence. This resistance can manifest through various mechanisms, including reduced drug accumulation within cancer cells, increased drug efflux, alterations in drug targets or cellular pathways, and many more.

#### 1.2.1. Drug Efflux Transporters

In OC, platinum-based compounds like cisplatin and paclitaxel represent standard chemotherapy options, but resistance to these drugs is common [22]. Reduced drug accumulation is a key mechanism behind cisplatin resistance in OC. This resistance occurs when cancer cells either limit the uptake of cisplatin or actively expel it, resulting in lower intracellular drug concentrations and diminished efficacy [23]. Efflux transporters like multidrug resistance protein 1 (MDR1) and multidrug resistance-associated protein 2 (MRP2) play pivotal roles by pumping cisplatin out of cancer cells, thereby reducing its cytotoxic impact [24]. Additionally, changes in copper transporters, such as copper transporter 1 (CTR1), can impede cisplatin’s entry into cells, contributing to resistance [25]. The epigenetic silencing of Otubain 2 (OTUB2) in OC leads to mitochondrial metabolic reprogramming by destabilizing sorting nexin 29 pseudogene 2 (SNX29P2), resulting in increased hypoxia-inducible factor-1 alpha (HIF-1α) levels and enhanced glycolysis through carbonic anhydrase 9 (CA9), contributing to tumor progression and chemoresistance [26]. P-glycoprotein (P-gp) is a crucial transporter protein that drives MDR in various cancers, including OC, by actively expelling chemotherapeutic drugs from cells, thereby reducing their intracellular concentrations and therapeutic effectiveness [27]. This process depends on energy from adenosine triphosphate (ATP) hydrolysis, making it a challenging obstacle in chemotherapy. The genetic variations P-gp, contribute to its differential expression and resistance mechanisms, impacting drug bioavailability and distribution. Beyond its role in MDR, P-gp, also known as MDR1, influences the tumor microenvironment (TME) and immune responses, with potential effects on dendritic cell function and T-cell activation [28]. Its influence on dendritic cells could alter antigen processing, presentation, maturation, or migration, while its effects on T-cell activation might involve changes in signaling pathways, cytokine environments, or cellular metabolism. These immunomodulatory actions could contribute to an immunosuppressive TME, affecting tumor infiltration by immune cells and potentially influencing the efficacy of immunotherapies. The dual roles of P-gp in drug resistance and immune modulation suggest intriguing possibilities for combination therapies that target both aspects, potentially enhancing anti-tumor immunity while combating chemoresistance [24,28].

#### 1.2.2. Cancer-Associated Fibroblasts (CAFs)

Cancer-associated fibroblasts (CAFs) are critical components of the TME in OC other than immune cells and normal epithelial cells, along with soluble factors, capillaries, and the extracellular matrix (ECM) [29]. CAFs, crucial in the reactive stroma, have significant interactions with cancer cells, impacting the TME by remodelling ECM and facilitating immune cell infiltration, tumor growth, survival, and chemoresistance [30]. Although CAFs are typically associated with promoting tumors, they display a lot of diversity and flexibility, with different subtypes such as myofibroblastic (myCAFs), inflammatory CAFs (iCAFs), and antigen-presenting CAFs (apCAFs) playing diverse and sometimes conflicting roles in the development of tumors, chemoresistance, cancer stem cell renewal, invasion, and immune cell polarization [31]—specifically, myCAFs (high α-SMA expression; involved in ECM production and remodeling), iCAFs (secrete cytokines like IL-6 and CXCL12; modulate immune response), apCAFs (exhibit antigen-presenting capabilities but inhibit T cell activation), and vascular CAFs (promote angiogenesis). While iCAFs, myCAFs, apCAFs, and vCAFs are well established, recent research has identified additional CAF subtypes, each characterized by unique markers and functions. Developmental CAFs (dCAFs) are implicated in developmental signaling pathways, influencing tumor progression and microenvironment remodeling [32]. CD146+ CAFs play roles in angiogenesis and the tumor vasculature, while α-SMA+ CAFs contribute to extracellular matrix (ECM) remodeling and contractility. Asporin+ and versican+ CAFs are involved in ECM modulation and composition [33]. FAP+ CAFs exhibit immunosuppressive functions, and PDGFR-α+ CAFs are associated with fibrotic processes [34]. Senescent CAFs promote tumor growth through the senescence-associated secretory phenotype (SASP) [35]. CD10+ and GPR77+ CAFs are linked to poor prognosis and chemoresistance, whereas podoplanin+ CAFs are involved in lymphangiogenesis and metastasis. These subtypes, while overlapping with established CAFs, exhibit distinct roles within the tumor microenvironment [36,37,38]. Specifically, the CAF_c1 subtype was identified in OC; this subtype is associated with unfavorable prognoses and resistance to immunotherapy. Its characterization suggests it plays a critical role in modulating the TME and impacting clinical outcomes [37]. In OC, CAFs play a significant role in tumor progression and resistance to treatment by activating various signaling pathways like TGF-β, Hedgehog, Notch, and Wnt. These pathways activate CAFs, interact with cancer cells, and reshape the tumor microenvironment. Signaling by TGF-β triggers CAF activation and supports fibrosis, which directly stimulates tumor cell growth, proliferation, invasion, and metastasis through various pathways like epidermal growth factor receptor (EGFR)/extracellular signal-regulated kinase (ERK)/phosphatidylinositol 3-kinase (PI3K)/protein kinase B (Akt)/mitogen-activated protein kinase (MAPK). In addition, the Hedgehog signaling pathway has a major role in the activation and functioning of CAFs in OC [38,39]. The activation of Hedgehog signaling in CAFs results in their differentiation and growth, which helps change the ECM and establish a conducive environment for cancer stem cells (CSCs). This activation increases the stiffness of the ECM, encourages the invasion and spread of cancer cells, and helps them withstand chemotherapy and targeted treatments by forming a shielded environment [40]. The Notch signaling pathway is largely activated by physical cell–cell contact between the signal-sending Notch ligand and the signal-receiving Notch receptor. This pathway is a widely preserved ligand–receptor system and has a crucial impact on multiple facets of OC biology, such as cancer stemness, angiogenesis, epithelial–mesenchymal transition, tumor immunity, and drug resistance [41]. CAFs also activate Wnt signaling, which plays a critical role by regulating ATP-binding cassette superfamily G member 2 (ABCG2) expression and promoting chemo-resistance via β-catenin activation and the upregulation of EMT transcription factors like SNAIL [42]. Lastly, interleukins, especially IL-6 and IL-8, aid in communication between CAFs and cancer cells, enhancing inflammation, angiogenesis, and immune evasion. They shape the immune landscape by recruiting and polarizing immune cells, often towards pro-tumoral phenotypes. CAFs modulate T cell responses through antigen presentation and cytokine secretion, which can either activate or suppress T cell function [43]. This eventually contributes to the formation of an immunosuppressive microenvironment that inhibits anti-tumor immunity [44]. CAFs can also influence the efficacy of immunotherapies by altering immune cell infiltration and function, creating both physical and chemical barriers that impede immune cell access to tumor cells. CAFs also alter the extracellular matrix using pathways that involve matrix metalloproteinases, which help cancer cells invade and spread to other parts of the body [45]. Furthermore, altered metabolism in CAFs, including increased glycolysis, aids in the sustenance of tumor proliferation and viability. These cells show metabolic reprogramming with higher glucose uptake and reactive oxygen species (ROS) production, which is connected to their differentiation. During periods of oxidative stress, CAFs have the ability to trigger autophagy, a mechanism that boosts the aggressiveness of cancer cells [46]. Different factors within the tumor microenvironment, such as inflammation, oxidative stress, DNA damage, and TGF-β signaling, impact the activation of CAFs. The intricate relationships and adjustments of CAFs are vital in influencing the tumor microenvironment and facilitating cancer progression and spread [47]. CAF-derived periostin (POSTN) and versican (VCAN) play significant roles in CAF activation and tumor progression. POSTN activates fibroblasts and promotes cancer cell migration and invasion through the PI3K/Akt pathway, while VCAN enhances cell motility and invasiveness via the NF-κB signaling pathway [48]. Furthermore, CAF-derived exosomes carrying miRNAs, proteins, and metabolites can transfer chemoresistance traits to cancer cells. These insights highlight the potential of targeting CAFs and their secreted factors, modulating immune responses, and inhibiting ECM remodeling as therapeutic strategies for improving outcomes in OC [29,49]. Table 1 displays the various types of CAFs involved in OC progression.

#### 1.2.3. Cell Death Pathways

Chemoresistance in OC is also strongly influenced by disruptions in apoptosis pathways, both intrinsic and extrinsic [50]. The intrinsic apoptosis pathway, also called the mitochondrial pathway, is an important process for programmed cell death triggered by internal cellular stress. This pathway is activated by different intracellular signals like DNA damage, oxidative stress, or metabolic imbalances [51]. The process progresses through a series of important stages: Initially, the outer layer of the mitochondria becomes permeable, resulting in the cytochrome c being released into the cytosol. The cytochrome c that is let go then connects with Apaf-1 and pro-caspase-9 to create the apoptosome complex. The creation of the apoptosome triggers caspase-9, causing the activation of caspase-3, resulting in cell death [52]. The Bcl-2 family of proteins, comprising pro-apoptotic (Bax, Bak, Bad, Bim) and anti-apoptotic (Bcl-2, Bcl-xL, Mcl-1) members, tightly controls the pathway. These proteins regulate the discharge of cytochrome c from the mitochondria. Additional regulation comes from inhibitors of apoptosis proteins (IAPs) like XIAP, survivin, and apollon, which can directly block caspase activity [53]. In chemoresistance, OC is frequently mediated through the upregulation of anti-apoptotic Bcl-2 family proteins, such as Bcl-2, Bcl-xL, and Mcl-1, which prevent cytochrome c release from mitochondria and inhibit apoptosome formation. Additionally, OC may downregulate pro-apoptotic proteins like Bax and Bak, impairing mitochondrial outer membrane permeabilization (MOMP) and reducing cytochrome c release [54]. Mutations in apoptosis regulatory proteins, such as Apaf-1 or caspases, and the overexpression of IAPs further contribute to resistance by blocking caspase activation and apoptotic cell death signaling [55]. The extrinsic pathway in OC involves the process that begins when signals from outside the cell interact with death receptors located on the cell membrane. Important components of this route in OC consist of tumor necrosis factor receptor 1 (TNFR1), FAS ligand (Fas-L), death receptor 4 (DR4), and DR5 [56]. The binding of ligands to these receptors triggers events leading to apoptosis. The binding of the Fas ligand to the Fas receptor triggers a signaling cascade. The interaction results in the death-inducing signaling complex (DISC) assembly, which involves recruiting the Fas-associated death domain (FADD) and pro-caspase-8. DISC formation activates caspase-8, which initiates the caspase cascade downstream; when activated, these pathways come together to activate effector caspases—specifically, caspase-3—that carry out the apoptotic process [57]. This consists of DNA fragmentation, membrane blebbing, and other characteristics of apoptotic cell death leading to apoptosis. Similar to the Fas pathway, when TRAIL binds to its receptors (TRAIL-R1 and TRAIL-R2), it triggers the formation of DISC. This initiation results in caspase-8 activation and then apoptotic signaling [58,59]. Occasionally, the extrinsic pathway may enhance the apoptotic signal by activating the intrinsic (mitochondrial) pathway via splitting the protein Bid by caspase-8, connecting both pathways. However, resistance to cell death through the extrinsic pathway can arise from various mechanisms. A common way involves decreasing levels or changing the structure of death receptors like Fas and TRAIL receptors on cancer cells, which decreases their capacity to start apoptosis when a ligand attaches. Moreover, cancer cells have the ability to increase the expression of anti-apoptotic proteins such as FLIP (FLICE-like inhibitory protein), which hinders caspase-8 activation at the DISC, thereby preventing further apoptotic signaling. A different method of resistance includes the disruption of downstream signaling pathways to hinder caspase activation or enhance pro-survival signals, allowing cell survival even when death receptors are activated [60,61,62]. In addition, changes in the tumor microenvironment, like cytokines or interactions with the surrounding tissue, can also play a role in making cancer cells less susceptible to death receptor-induced apoptosis by altering their sensitivity. Understanding these mechanisms opens up potential therapeutic strategies, including gene therapy, small molecule inhibitors, and combination therapy to counteract chemoresistance.

#### 1.2.4. DNA Damage

Deoxyribonucleic acid (DNA) repair pathways play a critical role in maintaining genome stability, but their overactivity in cancer cells can lead to resistance against chemotherapeutic agents [63]. Platinum-based chemotherapy, the standard treatment for OC, targets cancer cells by causing DNA damage, especially through DNA adducts. Cancer cells can develop resistance by enhancing DNA repair pathways, negating the effects of these chemotherapeutic agents [64]. Four main pathways involved in repairing DNA contribute to resistance to chemotherapy in ovarian cancer: homologous recombination (HR), non-homologous end joining (NHEJ), nucleotide excision repair (NER), and base excision repair (BER). HR plays a crucial role in DNA damage repair and is particularly significant in OC, especially in high-grade serous ovarian carcinoma (HGSOC) [65]. This DNA repair pathway, which involves key proteins such as BRCA1, BRCA2, and RAD51, is responsible for repairing double-strand breaks using a homologous template. In OC, defects in the HR pathway, often due to mutations in BRCA1/2 or other HR genes, contribute to genomic instability and cancer development [66]. However, these deficiencies also make tumors more susceptible to certain treatments. HR-deficient OC are generally more sensitive to platinum-based chemotherapy and PARP inhibitors, the latter exploiting synthetic lethality in HR-deficient cells [67,68]. The HR status of ovarian tumors has become a critical factor in treatment decisions, influencing the use of PARP inhibitors and informing prognosis. Meanwhile, NHEJ is a primary method for fixing double-strand breaks (DSBs) in DNA, functioning during all stages of the cell cycle but being especially crucial in the G0, G1, and early S phases. NHEJ does not need a homologous template for repair, unlike homologous recombination [69]. Non-homologous end joining connects damaged DNA ends without requiring matching sequences. It includes important proteins like Ku70/80, DNA-PKcs, XRCC4, DNA Ligase IV, and XLF. Increased NHEJ function may play a role in resisting DNA-damaging chemotherapies such as platinum drugs through the repair of drug-induced DSBs. In cases of OC with BRCA mutations or HR deficiency, the use of NHEJ for repairs may increase, which could impact how treatments work [70]. The equilibrium of NHEJ and HR can impact genomic stability and treatment outcomes in OC [71]. The NER pathway involves several key proteins, including ERCC1, XPA, XPB, and XPF, which work together to recognize and remove damaged DNA segments. In OC, the NER pathway’s ability to remove platinum-DNA adducts is a significant factor in chemoresistance. The overexpression or increased activity of NER proteins can lead to the enhanced repair of chemotherapy-induced DNA damage, allowing cancer cells to survive and continue proliferating despite treatment [72]. This is particularly relevant for platinum-based therapies like cisplatin and carboplatin, which are standard treatments for OC. Excision repair cross-complementation group 1 (ERCC1) is especially notable in this context. Its overexpression has been associated with poor prognosis and reduced sensitivity to platinum-based chemotherapy in OC patients. The ERCC1-XPF complex, which acts as an endonuclease in the NER pathway, is crucial for the excision step of repair. In the mechanism of action, ERCC1, combined with XPF/ERCC4, cuts damaged DNA at the beginning of NER, essential for eliminating DNA damage caused by treatments such as cisplatin, a common chemotherapy drug for OC [73]. Moreover, ERCC1 genetic variations are associated with cisplatin tolerance in patients with OC, emphasizing the significance of NER in resistance to therapy. Base excision repair (BER) is an essential DNA repair process that deals with minor base damage from OC. OGG1 variations, like Ser326Cys, are associated with higher vulnerability to OC because of their reduced ability to repair oxidative damage. Likewise, mutations in MUTYH play a part in the risk of OC, highlighting the importance of BER glycosylases in cancer susceptibility [74]. Increased levels of APE1 are linked to the aggressiveness of OC and a negative outcome, especially in later stages. Moreover, XRCC1 variations like Arg399Gln and Arg194Trp serve as a structural protein in BER, aiding in the connection between repair enzymes such as PARP1, LIG3, and POLB, and eventually leading to higher susceptibility to OC, and they are linked to poorer clinical results such as resistance to platinum-based treatments. Overall, BER is essential for preserving genomic stability in OC and has implications for cancer vulnerability, outlook, and treatment reactions, especially with new targeted therapies [75]. Targeting these pathways offers potential strategies for overcoming chemoresistance. PARP inhibitors are effective in HR-deficient cancers, particularly BRCA1/2-mutated ones, by inhibiting PARP1 and blocking DNA repair processes [76,77]. Using siRNA or shRNA to target RAD51 and BRCA1/2 can enhance chemotherapy sensitivity. DNA-PK inhibitors reduce repair efficiency in the NHEJ pathway, while siRNA targeting RIF1 enhances cisplatin sensitivity. In the NER pathway, siRNA targeting the ERCC1-XPF complex increases platinum drug sensitivity. Additionally, in the BER pathway, siRNA targeting XRCC1 and investigating DNA polymerase β (Polβ) as a therapeutic target could provide effective strategies for combatting chemoresistance [78,79].

#### 1.2.5. Cancer Stem Cells (CSCs)

Cancer stem cells (CSCs) within OC are a key driver of chemoresistance, tumor relapse, and metastasis [80,81]. These cells possess unique properties like self-renewal, differentiation, and high tumorigenic potential, making them particularly resistant to conventional chemotherapy [82]. CSCs can be identified by specific markers such as CD24, CD44, CD117, CD133, and aldehyde dehydrogenase (ALDH) [83]. Their resistance mechanisms include slow proliferation, a high expression of ATP transporters, and the ability to inactivate cell death pathways. Critical transcription factors like SOX2, OCT4, and NANOG play a pivotal role in maintaining stem-like characteristics, and their overexpression correlates with chemoresistance [84], along with heightened levels of surface markers linked to stem cells (such as CD44 and CD117). These cells exhibit stem markers like ALDH1, LGR5, LEF1, CD133, and CK6B, display enhanced sphere formation, and demonstrate an increased potential for transformation after Trp53 and Rb1, two commonly mutated tumor-suppressor genes in HGSOC, are deactivated. Furthermore, stem-like side population cells have been discovered in mouse OC [85]. These cells have increased levels of membrane ABC transporters, which help remove chemotherapeutic drugs, causing resistance to treatment. EMT in OC is also strongly linked to CSCs, explaining why metastases often show resistance to treatment. In OC, Snail and Slug EMT, transcription factors help cells gain stemness markers and turn off the p53-mediated cell death process [86]. WNT/β-catenin signaling plays a crucial role in CSCs and the development of OC, controlling cell growth, apoptosis, and the transition of cells during EMT. Alterations in proteins of the WNT pathway contribute to the development of OC and its resistance to treatment. Unlike many solid tumors, OC tends to spread mainly to the peritoneal cavity due to the absence of anatomical barriers [87]. This dissemination occurs via individual cells or clusters of cancer and stromal cells drifting in ascites fluid, leading to numerous metastatic tumors. OC cells often favor the omentum, a peritoneal fold rich in adipose tissue [9]. OC cell attachment to the peritoneal mesothelium involves the OCSC markers CD44 and integrin-β1, which bind to the HA receptor on mesothelial cells. The knockdown of CSC factors with shRNA resensitizes ovarian cancer cells to chemotherapy, indicating their potential as therapeutic targets. The Janus kinase-signal transducer and activator of transcription (JAK-STAT) pathway is also implicated in stemness and tumorigenicity [88]. The inhibition of this pathway with ruxolitinib, an FDA-approved JAK1/2 inhibitor, has shown promise in reducing chemoresistance and increasing the effectiveness of chemotherapeutics like paclitaxel. Combination therapy targeting both CSCs and conventional cancer cells might be a more effective approach to overcoming chemoresistance and reducing tumor recurrence in OC [89].

#### 1.2.6. Immune Evasion

Another significant challenge in OC treatment is that immune evasion poses a major obstacle in treating ovarian cancer due to the tumor’s creation of a diverse and intricate environment that interferes with anti-tumor immune reactions, reducing the efficacy of immunotherapy when compared to other cancers [17]. The immunosuppressive setting is controlled by different cellular elements and processes involving Tregs migrating into OC primarily due to the action of C-C motif chemokine 22 (CCL22) in the tumor microenvironment. In OC lesions, Tregs suppress the immune response to tumor-associated antigens by inhibiting the secretion of interferon-gamma (IFN-γ), interleukin-2 (IL-2), IL-10, and TGF-β by effector T cells [90]. TGF-β aids in differentiating naïve T cells into Tregs, which contribute to immune suppression and promote EMT along with angiogenesis with the upregulation of IL-8. This suppression makes Tregs a significant prognostic indicator in ovarian cancer patients [91]. Patients with ovarian tumors that contain Tregs have a higher hazard ratio for death, and an increased number of Tregs within tumors is associated with more aggressive cancer forms; other studies have mentioned that tumor-related macrophages, especially those that are M2-polarized, inhibit immune responses by releasing immunosuppressive substances and facilitating Treg recruitment. Plasmacytoid dendritic cells (pDCs) contribute to the generation of CD8+ Tregs and support angiogenesis, whereas myeloid-derived suppressor cells (MDSCs) inhibit immune reactions via molecules such as arginase and indoleamine-2,3-dioxygenase (IDO) and recruit additional Tregs to enhance the immunosuppressive environment [92]. The tumor cells play a role in creating this environment by reducing MHC class I expression, producing immune checkpoint molecules, and releasing immunosuppressive factors. Moreover, stromal cells, metabolic factors, extracellular vesicles, as well as a modified cytokine and chemokine environment contribute to the development and upkeep of this intricate immunosuppressive environment [93]. Comprehending these complex mechanisms is essential for creating focused immunotherapies to tackle immune evasion in OC, with current tactics investigating checkpoint inhibitors, combination therapies, CAR-T cell methods, targeting the tumor microenvironment, vaccine strategies, and modulating metabolism. Researchers are working to enhance the effectiveness of immunotherapy in treating ovarian cancer by targeting various immune evasion mechanisms, with the goal of improving patient outcomes for this difficult disease.

#### 1.2.7. Extracellular Vesicles (EVs)

Recent studies have also demonstrated that extracellular vesicles (EVs) are involved in OC chemoresistance, as drug-resistant cells release a higher number of EVs and pass on their resistance to susceptible cells [94]. EVs from OC cells treated with cisplatin contribute to resistance and invasiveness. They carry proteins like plasma gelsolin, P-glycoprotein, GATA3, and STAT3, which are important for drug resistance [95]. In particular, plasma gelsolin promotes resistance to CDDP by inducing CD8 T-cell death and increasing GSH levels. High levels of plasma gelsolin in EVs are linked to poor outcomes and platinum resistance. GATA3 in ascites triggers drug resistance in ovarian cancer cells [96]. EVs with P-glycoprotein transfer it between cells, boosting resistance to drugs like adriamycin and paclitaxel. STAT3-containing EVs enhance the resistance and migration of OC cells by activating genes involved in drug resistance and invasion. Moreover, EVs contain miRNAs that play a role in OC chemoresistance, underlining the significance of EVs in creating new treatment approaches [97]. For instance, miR-21-5p enhances carboplatin resistance by activating glycolysis and increasing the expression of the ABC family and detoxification enzymes. On the other hand, miR-891-5p increases platinum resistance by affecting DNA repair mechanisms. miR-21 carried by CAF- and CAA-derived EVs reduces sensitivity to paclitaxel, while miR-98-5p carried by CAF-derived EVs promotes cisplatin resistance [98,99]. Additionally, miR-30a-5p inhibits SOX9 expression, reducing resistance in SKOV3 cells. Other notable EV miRNAs, such as miR-223, miR-1246, and miR-433, also contribute to chemoresistance through various mechanisms. EV miRNAs like miR-98-5p, miR-130a, and miR-891-5p are associated with platinum resistance, while miR-100, miR-143, and miR-30a-5p can resensitize OC cells to platinum-based treatments [100]. Exosomal long non-coding RNA (lncRNA) activated by TGF-β (ATB) from OC cells enhances the TME by increasing the viability, angiogenesis, and migration of HUVECs. This promotes tumorigenesis through the miR-204-3p/TGFβR2 axis [101]. The knockdown of lncRNA ATB inhibits ovarian cancer growth and angiogenesis, offering potential therapeutic insights. PLADE, a lncRNA in ascites-derived exosomes, is studied in HGSOC for its role in improving cisplatin sensitivity. Reduced PLADE expression is linked to resistance in patients and metastatic tumors. Higher levels are linked to longer survival. PLADE inhibits cell growth and migration while promoting cell death. It binds to heterogeneous nuclear ribonucleoprotein D (HNRNPD), reducing its levels through VHL-mediated ubiquitination and increasing cisplatin sensitivity [102]. Similarly, elevated exosomal SOX2-OT in the plasma of OC patients promotes migration, invasion, and proliferation while inhibiting apoptosis in cancer cells by sponging miR-181b-5p targeting SCD1. The overexpression of SCD1 and miR-181b-5p inhibition reverse these effects. The depletion of exosomal SOX2-OT reduces tumor growth in vivo, indicating its role in enhancing ovarian cancer malignancy through the miR-181b-5p/SCD1 axis, offering potential therapeutic targets [103]. Additionally, EVs play a critical role in immune evasion in OC by transporting immune checkpoint molecules such as PD-L1 to suppress T-cell activation and inducing dendritic cell apoptosis through molecules like FasL. They also drive macrophage polarization towards the immune-suppressive M2 phenotype, which is associated with tumor progression. Additionally, EVs from OC cells hinder antigen presentation by blocking dendritic cell activation [104]. Moreover, these EVs reduce the cytotoxicity of CD4+ and CD8+ T cells and natural killer (NK) cells against tumor cells, thus facilitating immune evasion. They regulate T-cell responses by inhibiting receptor signaling and cytokine release, thereby impairing T-cell proliferation and activation [105]. Furthermore, EVs contribute to an immunosuppressive tumor microenvironment by promoting regulatory T cell maturation and inhibiting B-cell function. Despite their role in immune evasion, EVs also offer therapeutic potential. They can stimulate cytokine release from monocytes and provide strategies for modulating macrophage behavior and enhancing antigen presentation, which could be leveraged for immunotherapy [106]. Understanding the intricate role of EVs in immune evasion in OC is crucial for developing effective treatments. Further investigation into EV cargoes and their interactions within the tumor microenvironment will be essential to optimize treatment strategies and improve patient outcomes. Research has identified various EV cargos associated with OC metastasis, including proteins and miRNAs from OC cells like SKOV3. Specific miRNAs like miR-105, miR-6126, and miR-7 influence proliferation and metastasis. EVs from human adipose mesenchymal stem cells inhibit OC cell proliferation [107]. Additionally, EVs are crucial in angiogenesis, promoting tumor development via molecules like VEGF and miR-141-3p [108]. EVs also mediate metastasis by altering the tumor microenvironment and promoting epithelial–mesenchymal transition through molecules like miR-205 and LIN28. These findings highlight the diverse mechanisms by which EVs contribute to OC metastasis [109]. CSCs play a crucial role in tumor progression and therapy resistance, including in OC, where they survive chemotherapy and lead to drug-resistant metastases. EVs contribute to this process by promoting CSC characteristics and facilitating metastasis through miR-454 transfer [110]. EVs are key players in the regulation of stem cell behavior. For instance, EVs derived from OC cells can transfer factors that maintain the stemness and self-renewal capacity of CSCs, such as Wnt and Notch signaling molecules. These signaling pathways are crucial for the maintenance of CSC characteristics, contributing to tumor growth, metastasis, and resistance to conventional therapies. EVs can influence the differentiation state of cells. In some cases, they may promote the dedifferentiation of cancer cells into a more stem-like state, contributing to tumor heterogeneity and treatment resistance [111]. EVs contribute to the modulation of DDR in OC by transporting DNA repair enzymes, miRNAs, and other regulatory molecules. For example, EVs from ovarian cancer cells have been found to contain proteins involved in the BER pathway, such as APE1 and POLB, involved in the base excision repair pathway, as well as regulatory miRNAs, which can be transferred to recipient cells to enhance their DNA repair capacity. This transfer can promote the survival of cancer cells by improving their ability to repair DNA damage, thereby contributing to tumor progression and resistance to therapy [112]. In addition, CAFs, in turn, secrete EVs that contain pro-tumorigenic molecules, such as TGF-β and IL-6. These factors are known to enhance cancer cell proliferation, invasion, and metastasis. The EV-mediated transfer of these molecules can activate signaling pathways in ovarian cancer cells, promoting aggressive tumor behavior and resistance to therapy. For instance, TGF-β carried by CAF-derived EVs can activate the epithelial–mesenchymal transition (EMT) in cancer cells, a process that facilitates metastasis and contributes to the formation of secondary tumors [113]. EVs derived from OC cells can also transfer miRNAs, such as miR-21, which target and downregulate pro-apoptotic genes like PTEN, thereby promoting cell survival and resistance to chemotherapy. The delivery of miR-21 through EVs can activate signaling pathways that inhibit apoptosis and enhance cancer cell proliferation, contributing to tumor growth and chemoresistance [114]. EVs can also influence the TME by promoting the transformation of stromal cells into CAFs. Examination was carried out on the transfer of H19 from tumor cells to CAFs by exosomes, revealing that exosomes from tumors could increase the levels of LOXL2 and COL1A1, supporting the proliferation, activation, and migration of CAFs. Animal experiments supported these results by showing elevated levels of LOXL2 and COL1A1 and reduced levels of miR-29c-3p in mixed xenograft tumors, enhancing their migratory and proliferative abilities. Research into EV cargoes and their interactions within the tumor microenvironment is crucial for developing effective treatments [115]. Table 2 displays the roles of EVs in mediating chemoresistance in OC. Table 3 summarizes the key mechanisms contributing to chemoresistance in OC. Figure 2 shows the immunosuppressive microenvironment leading to chemoresistance involving EV, CAFs, and CSCs.

## 2. Extracellular Vesicles as Emerging Drug Delivery

EVs are naturally occurring nanovesicles released by various cell types, including reticulocytes, mesenchymal stem cells, T cells, B lymphocytes, NK cells, dendritic cells, and some tumors [116]. EVs could indeed be helpful in overcoming the limitations of traditional nanoparticle-based drug delivery systems (NDDSs) such as liposomes. EVs are natural lipid bilayer vesicles released by cells and found in various body fluids such as blood, urine, saliva, and cerebrospinal fluid. EVs are heterogeneous in size and content, including exosomes, microvesicles, and apoptotic bodies, each with distinct biogenesis pathways and functions [117,118]. They are involved in physiological processes such as immune response modulation, tissue regeneration, and maintaining homeostasis. EVs also have implications in disease states, including cancer, neurodegenerative disorders, and cardiovascular diseases, making them a focus of research for diagnostic and therapeutic applications [117]. EVs can carry bioactive molecules and exhibit low toxicity, high biocompatibility, and inherent targeting capabilities due to their natural origin. In addition, EVs shield nucleic acids and other large biological molecules from degradation by enzymes and have the ability to pass through the blood–brain barrier to transport functional gene therapies to specific locations. These features make EVs promising drug delivery vehicles, offering potential solutions to the challenges of poor bioavailability, non-targeted delivery, and systemic toxicity associated with traditional NDDSs [119].

In detail, EVs are tiny particles enclosed by a membrane that is discharged by cells, such as exosomes (30–150 nm), microvesicles (100–1000 nm), and apoptotic bodies. Exosomes are intraluminal vesicles that begin with the internal budding of endosomes, forming intraluminal vesicles (ILVs). These ILVs mature into late endosomes, known as multivesicular bodies (MVBs), and eventually merge with the plasma membrane and release ILVs into the extracellular space as the exosome fusion of MVBs with the plasma membrane. Exosomes typically have a “cup-like” structure. Their main molecular components include cell-derived lipids, glycoconjugates, proteins, and nucleic acids (such as lncRNA, microRNA, and DNA) [120,121]. Exosomes contain an evolutionarily conserved set of proteins, including fusion and trafficking proteins (Rab2, Rab7, flotillin, and annexin), heat shock proteins (HSP70 and HSP90), integrins, tetraspanins (CD63, CD81, and CD82), cytoskeletal proteins (β-actin and tubulin), and synthesis proteins (Alix and Tsg101), some of which serve as specific markers for identifying exosomes [122]. 9. Contrarily, ectosomes, or microvesicles, are produced by budding and shedding from the plasma membrane [122,123]

The EV biogenesis involves diverse mechanisms: exosomes form within multivesicular bodies through ESCRT-dependent and independent pathways, ectosomes bud directly from the plasma membrane, and apoptotic bodies result from cellular fragmentation during programmed cell death. The recently identified exomeres are smaller, non-membrane-bound vesicles. The tetraspanin protein family is key in EV biogenesis, with differing exosome and ectosome formation functions. Various pathways result in EVs with distinct sizes, compositions, and functions, contributing to the complexity of extracellular communication [124,125]. Once released, EVs are internalized by target cells through multiple mechanisms, including direct membrane fusion and both clathrin-dependent and -independent endocytosis. Clathrin-independent pathways include caveolin-mediated uptake, macropinocytosis, phagocytosis, and lipid raft-mediated internalization. Key molecules facilitating uptake include PSGL-1, phosphatidylserine, AP2, and TIM4. These diverse biogenesis and uptake mechanisms allow for versatile EV-mediated signaling across different cell types and physiological contexts [123]. Understanding these processes is crucial for developing EV-based diagnostics and therapeutics, highlighting the importance of ongoing research in this field.

Exosomes are effective carriers for natural substances and chemotherapeutic drugs due to their small size and ability to penetrate cell membranes. Researchers have developed various methods for loading cargo into EVs, including electroporation, sonication, dialysis, co-precipitation, fusion with liposomes, microfluidics, chemical modification, and electrostatic interaction. While these methods have shown promise, they often require optimization for specific cargoes and EV types. For instance, melanoma-derived exosomes have been shown to be home to sentinel lymph nodes, demonstrating their intrinsic homing capabilities [126]. Exosomes loaded with anticancer drugs have already shown promise as a novel therapeutic approach in animal models. Studies have shown that exosomes can deliver curcumin effectively into cancer cells, enhancing its anticancer effects compared to curcumin alone, which suffers from low bioavailability [127]. Paclitaxel-loaded exosomes demonstrate enhanced tumor targeting and can potentially cross the blood–brain barrier. Cisplatin delivered via milk-derived exosomes can avoid drug resistance mechanisms [128]. Doxorubicin-loaded exosomes show reduced cardiotoxicity and improved targeting to specific cancer cells when modified with antibodies or targeting ligands [129]. Similarly, EV-mediated delivery systems have shown promise in increasing the availability and effectiveness of medications in OC [130]. This is especially important in OC, where resistance to drugs and toxicity throughout the body frequently present major obstacles to successful treatment. EVs containing therapeutic agents have the potential to target cancer cells effectively, overcoming drug resistance in ovarian cancer and reducing the systemic toxicity of traditional chemotherapy [131]. This focused strategy may result in better therapies with reduced side effects, potentially enhancing results for individuals with OC. Studies have demonstrated the efficacy of exosomes in delivering chemotherapeutics like doxorubicin, cisplatin, and paclitaxel, as well as natural compounds such as berry anthocyanidins, all showing enhanced cytotoxicity and anticancer activity in ovarian cancer models [132]. These approaches leverage exosomes’ unique properties, including their ability to cross biological barriers, natural targeting capabilities, and potential for reduced side effects compared to traditional drug delivery methods [133]. While these studies show promise, further research is needed to optimize exosome production, loading efficiency, and targeting, as well as to evaluate safety and efficacy in clinical trials before widespread application in ovarian cancer treatment. An investigation has mentioned that doxorubicin (DOX) nanoparticles (DN)@orange-derived EV (OEV) exhibit potent antitumor effects in vivo, significantly suppressing tumor growth compared to free DOX or DN. Immunohistochemical analysis suggests that DN@OEV inhibits OC proliferation and angiogenesis, contributing to its therapeutic efficacy [134]. A study demonstrates that Amla extract (AE) treatment leads to increased levels of miR-375 in exosomes secreted by SKOV3 cells of the OC cell line. This indicates that AE may modulate the tumor microenvironment by altering the composition of exosomal miRNAs, potentially affecting neighboring cells or distant sites [135]. Meanwhile, shikonin (SK) reduces the population of M2 macrophages in OC by suppressing exosome production and blocking exosomal GAL3-mediated β-catenin activation. This suggests that SK may have potential as a therapeutic agent for OC by modulating the tumor microenvironment and inhibiting tumor-promoting immune responses [136]. Zhang et al. [137] reported that curcumin can restore LncRNA MEG3 levels in EV from cisplatin-resistant OC cells. The upregulation of MEG3 reduced the expression of miR-214 in both cells and EV, thereby decreasing cell survival and cisplatin resistance. A study has mentioned that milk-based exosomes as delivery vehicles greatly increase the effectiveness of anthocyanidins (Anthos) found in berries, which has improved cisplatin’s effectiveness and decreased drug resistance in OC. The use of Anthos and paclitaxel in combination therapy delivered through exosomes shows synergistic antitumor effects, offering better treatment for OC [138]. EVs offer a promising and versatile platform for delivering chemotherapeutic drugs and natural compounds, enhancing their therapeutic efficacy while minimizing systemic toxicity. The successful application of EVs in ovarian cancer treatment opens new avenues for incorporating other natural compounds with known anticancer properties [139]. These compounds, such as resveratrol, epigallocatechin gallate (EGCG), and other polyphenols, could be explored for their combined effects with EVs, potentially leading to novel and more effective therapeutic strategies. Recent research has focused on MSC-derived products, particularly EVs, as a form of cell-free therapy. EVs, which carry various bioactive molecules, can be engineered to deliver therapeutic agents to tumor sites. This approach leverages the natural targeting and delivery capabilities of EVs, offering a promising alternative to direct MSC-based therapies. They offer several advantages over direct stem cell therapies, including a reduced risk of immune reactions, enhanced safety, and improved targeting precision. By modifying EV surfaces with targeting ligands and employing controlled release mechanisms, these vesicles can be tailored to increase the therapeutic impact while minimizing off-target effects and side effects [140,141]. Table 4 highlights the various drugs and natural compounds delivered by EVs, their sources, and the benefits they offer in improving OC therapy.

## 3. Challenges of EV as Drug Delivery Agents

Exosomes show great potential as carriers for cancer treatment drugs, yet their use in clinical settings is hindered by various obstacles. Despite their potential, challenges such as standardization, scalability, and regulatory considerations remain. These challenges are compounded by the complexity of isolating and producing EVs in sufficient quantities while maintaining their therapeutic properties [142,143]. Current isolation methods face several challenges: traditional methods often produce EVs in small quantities, which are insufficient for therapeutic applications, necessitating the development of large-scale production techniques to meet clinical demand [144]. Additionally, exosomes and other EVs can vary significantly in their composition, affecting their therapeutic efficacy and consistency, which poses a challenge for standardizing EV-based treatments. EVs are also quickly cleared from the bloodstream by the body’s reticuloendothelial system, leading to a short in vivo half-life, thereby reducing the time available for EVs to reach and affect their target cells [145]. Furthermore, current methods for loading therapeutic agents into EVs can compromise their integrity and reduce their effectiveness, making efficient and gentle loading techniques necessary to preserve the functional properties of EVs while ensuring they carry an adequate therapeutic payload [146]. Off-target effects, a limited understanding of delivery route complexity, and difficulties in monitoring and tracking the fate of EVs in the body after administration further complicate the development of EV-based drug delivery systems [147]. In order to bring the therapeutic potential of exosomes into clinical practice, overcoming these challenges will be crucial as the field advances. Continued research is essential for optimizing the production, loading efficiency, and targeting of EVs and for thoroughly evaluating their safety and efficacy. The future of EV-based drug delivery holds significant promise for improving outcomes in ovarian cancer and other challenging diseases, offering targeted and personalized therapeutic options with potentially fewer side effects compared to traditional treatments.

## 4. Future Directions

EVs have increasingly been recognized for their potential in drug delivery, leveraging their inherent properties to enhance therapeutic efficacy and precision. Established uses of EVs in drug delivery include their role in nucleic acid delivery, where they transport therapeutic RNAs, such as small interfering RNAs (siRNAs) and microRNAs (miRNAs), effectively protecting these sensitive molecules from degradation and facilitating their uptake by target cells [148]. In cancer therapy, EVs have been employed to deliver chemotherapeutic agents and targeted therapies directly to tumor cells, improving drug accumulation at the tumor site while minimizing systemic toxicity. This approach has been exemplified by the use of EVs loaded with drugs like paclitaxel or doxorubicin, which have shown enhanced therapeutic efficacy in preclinical cancer models [149]. Additionally, EVs are utilized for immunomodulation, carrying cytokines, antigens, or immune checkpoint inhibitors to modulate immune responses, thereby contributing to the field of cancer immunotherapy [150]. Recent advancements in EV-based drug delivery have further expanded their potential applications. Innovations in surface engineering have enabled the modification of EV surfaces with targeting ligands or peptides, enhancing their specificity for particular cells or tissues and improving targeted delivery while reducing off-target effects [151]. Advances in controlled release systems allow EVs to release their therapeutic cargo in response to specific stimuli, such as changes in pH or enzymatic activity, providing a more precise and controlled delivery of therapeutics [152]. Improving isolation and characterization techniques for EVs is crucial for advancing both research and clinical applications. Current isolation methods face challenges with the purity, yield, and reproducibility. Enhancements include size-based isolation techniques like tangential flow filtration (TFF) and asymmetric flow field-flow fractionation (AF4) for better separation, affinity-based isolation with more specific antibodies or aptamers targeting EV surface markers, and microfluidic devices that efficiently isolate EVs with minimal sample volumes [153,154]. Refining polymer-based precipitation methods such as polyethylene glycol (PEG) precipitation and optimizing density gradient ultracentrifugation protocols to separate EV subpopulations are also essential. For characterization, advanced microscopy techniques like STORM and PALM offer the nanoscale resolution of EV structures, while single-vesicle analysis methods such as nanoflow cytometry and Raman spectroscopy enable the detailed analysis of individual EVs. Sensitive mass spectrometry techniques enhance proteomics and lipidomics profiling, and next-generation RNA sequencing methods expand the characterization of EV RNA content, including long non-coding and circular RNAs. Functional assays for assessing EV biological activity, surface plasmon resonance for studying EV–protein interactions, and cryo-electron tomography for detailed 3D reconstructions of EV structures further improve our understanding. Standardizing these methods across the field will enhance the accuracy, reproducibility, and comparability of EV research and its applications [155,156,157]. Furthermore, the exploration of combination therapies involving EVs, such as using EVs to deliver gene-editing tools like CRISPR/Cas9 or integrating EVs with radiation therapy, represents a novel approach to enhancing overall treatment efficacy [158,159]. These advancements underscore the growing versatility and potential of EVs as drug delivery vehicles, offering promising avenues for the development of targeted and effective therapies across various medical fields. Figure 3 shows the overall summary of the potential of EVs as drug delivery vehicles.

## 5. Conclusions

The field of OC treatment is at a critical juncture, facing significant challenges in overcoming chemoresistance while exploring innovative therapeutic approaches. The complex mechanisms of drug resistance, including drug efflux, apoptosis disruption, enhanced DNA repair, and the influential roles of cancer stem cells and the tumor microenvironment, underscore the need for multifaceted treatment strategies. EVs have emerged as a promising platform for drug delivery in OC, offering unique advantages such as low toxicity, high biocompatibility, and inherent targeting capabilities. The potential of EVs to deliver both conventional chemotherapeutics and natural compounds effectively to cancer cells while minimizing systemic toxicity represents a significant advancement in the field. However, translating EV-based therapies from preclinical studies to clinical application faces several hurdles. These include optimizing EV production and drug loading methods, addressing heterogeneity in EV composition, and overcoming rapid clearance from circulation. Resolving these challenges will be crucial for realizing the full therapeutic potential of EVs in OC treatment. As research progresses, the integration of EVs with other emerging technologies and the exploration of novel drug combinations may lead to more personalized and effective treatment strategies. The future of OC therapy lies in harnessing the unique properties of EVs while continually refining our understanding of chemoresistance mechanisms. This approach holds promise for improving treatment outcomes and quality of life for patients with this challenging malignancy. Ultimately, the successful development of EV-based therapies could revolutionize OC treatment, offering new hope to patients and potentially extending to other difficult-to-treat cancers. Continued investment in research and clinical trials will be essential for translating these promising findings into tangible benefits for patients.

## Figures and Tables

**Figure 1 biomedicines-12-01806-f001:**
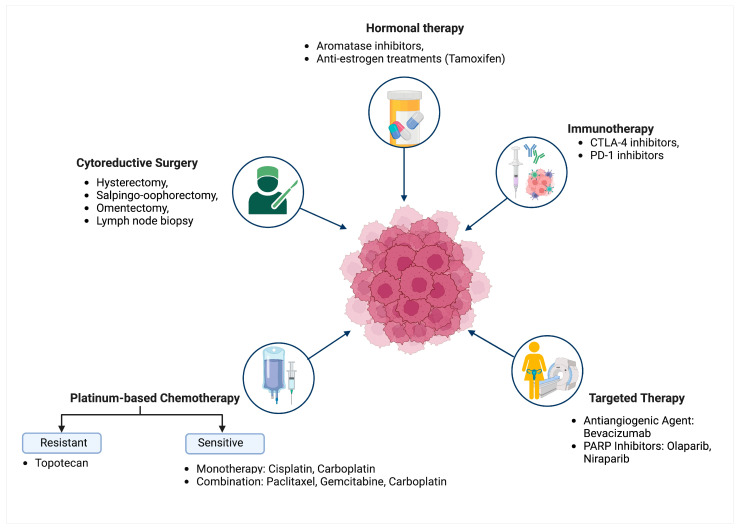
Advanced strategies in OC treatment. Initial Treatment for OC involves cytoreductive surgery and platinum-based chemotherapy to remove tumor tissue and target residual cancer cells. Recurrent OC management varies based on platinum sensitivity, utilizing combination chemotherapy or alternative agents like topotecan. Targeted Therapies include antiangiogenic agents, PARP inhibitors, and immune checkpoint inhibitors for extended progression-free survival. Hormone Therapy explores estrogen and progesterone dependencies with aromatase inhibitors and anti-estrogen treatments for OC treatment options.

**Figure 2 biomedicines-12-01806-f002:**
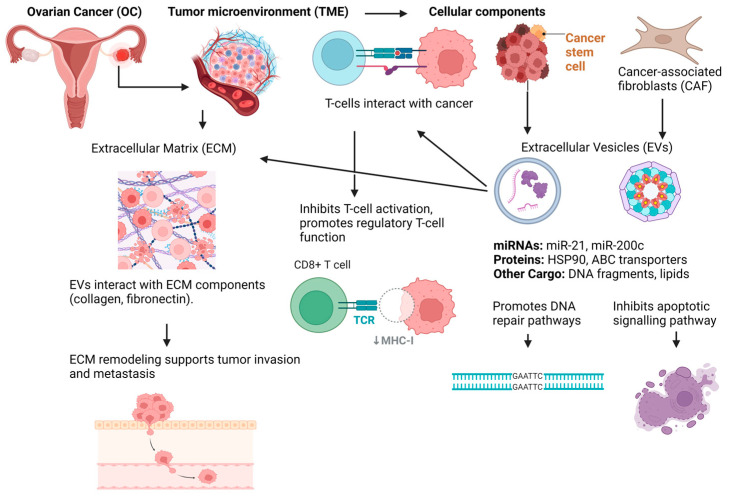
The chemoresistance mechanisms in OC focusing on EV and their interactions with the TME. Cancer cells and CAFs release EVs into the extracellular space. EV contents, including miRNAs (e.g., miR-21, miR-200c), proteins (e.g., HSP90), ABC transporters, and other cargo (e.g., DNA fragments, lipid droplets), are taken up by neighboring cancer cells and distant metastatic sites. The EV cargo promotes DNA repair and inhibits apoptosis, contributing to chemoresistance. TME-mediated communication has interactions involving the ECM and EVs, leading to invasion and metastasis.

**Figure 3 biomedicines-12-01806-f003:**
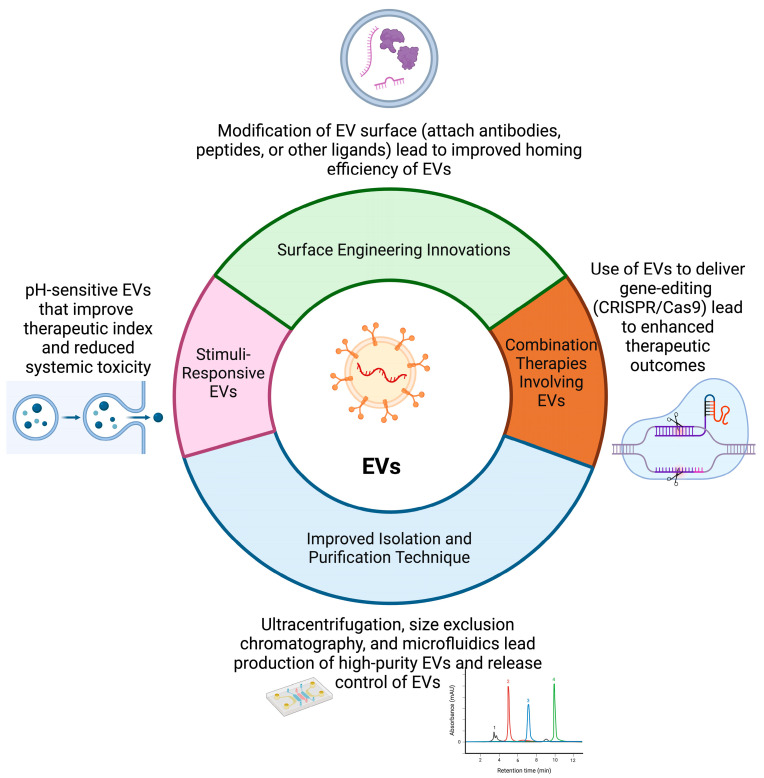
Advancements highlight the growing versatility and potential of EVs as drug delivery vehicles, offering promising avenues for targeted and effective therapies across various medical fields.

**Table 1 biomedicines-12-01806-t001:** CAF Subtypes and their implications for OC.

CAF Subtype	Key Characteristics	Functions	Implications for OC
Myofibroblastic CAFs (myCAFs)	High α-SMA expression	- ECM production and remodeling - Alter tumor stiffness	- Facilitate cancer cell invasion
Inflammatory CAFs (iCAFs)	Secrete cytokines (e.g., IL-6, CXCL12)	- Modulate immune response - Promote inflammation	- Affect tumor progression
Antigen-presenting CAFs (apCAFs)	Exhibit antigen-presenting capabilities	- Interact with T cells - Inhibit T cell activation	- Contribute to immune evasion
Vascular CAFs (vCAFs)	Promote angiogenesis	- Support tumor vasculature - Enhances tumor growth	- Facilitate metastasis
Developmental CAFs (dCAFs)	Involved in developmental signaling	- Influence tumor progression - Microenvironment remodeling	- Affect tumor development - Alter tumor–stroma interactions
CD146+ CAFs	Associated with angiogenesis	- Promote tumor vasculature - Support tumor growth	- Enhance metastatic potential
α-SMA+ CAFs	Contribute to ECM remodeling	- Enhance contractility - Increase tumor stiffness	- Promote cancer cell invasion
Asporin+ and Versican+ CAFs	Involved in ECM modulation	- Alter ECM composition - Affect the tumor microenvironment	- Influence cancer cell behavior
FAP+ CAFs	Exhibit immunosuppressive functions	- Suppress anti-tumor immunity - Promote immune evasion	- Reduce immunotherapy efficacy
PDGFR-α+ CAFs	Associated with fibrotic processes	- Promote fibrosis - Alter tumor stroma	- May affect drug penetration
Senescent CAFs	Exhibit senescence-associated secretory phenotype (SASP)	- Promote tumor growth - Enhance tumor progression	- Affect the treatment response
CD10+ and GPR77+ CAFs	Linked to poor prognosis	- Promote chemoresistance - Reduce treatment efficacy	- Worsen patient outcomes
Podoplanin+ CAFs	Involved in lymphangiogenesis	- Promote metastasis - Enhance tumor spread	- Worsen prognosis
CAF_c1 subtype	Associated with unfavorable prognoses	- Promote resistance to immunotherapy - Reduce the efficacy of immune-based treatments	- Worsen patient outcomes

**Table 2 biomedicines-12-01806-t002:** Chemoresistance in OC related to EV.

EV Mechanisms	Description	Key Players	Impact on Chemoresistance	References
EV Release and Drug Resistance Transfer	Drug-resistant OC cells release more EVs, passing on resistance to susceptible cells.	Plasma gelsolin, P-glycoprotein, GATA3, STAT3	Promotes resistance to cisplatin (CDDP) by inducing CD8 T-cell death, increasing GSH levels, and enhancing drug resistance and invasiveness	[94,95,96]
EV Proteins and Chemoresistance	EVs carry proteins important for drug resistance.	Plasma gelsolin, GATA3, P-glycoprotein, STAT3	Plasma gelsolin promotes CDDP resistance, GATA3 in ascites induces drug resistance, P-glycoprotein in EVs boosts resistance to adriamycin and paclitaxel, and STAT3 in EVs enhances resistance and migration	[94,95,96]
miRNAs in EVs	EV miRNAs contribute to OC chemoresistance.	miR-21-5p, miR-891-5p, miR-98-5p, miR-30a-5p, miR-223, miR-1246, miR-433	miR-21-5p enhances carboplatin resistance, miR-891-5p increases platinum resistance, miR-98-5p promotes cisplatin resistance, and miR-30a-5p reduces resistance	[97,98,99,100]
lncRNAs in EVs	EV lncRNAs play roles in chemoresistance and tumor progression.	lncRNA ATB, PLADE, SOX2-OT	lncRNA ATB enhances TME, PLADE improves cisplatin sensitivity, and SOX2-OT promotes migration, invasion, and proliferation	[101,102,103]
Immune Evasion via EVs	EVs transport immune checkpoint molecules and modulate immune responses.	PD-L1, FasL	Suppresses T-cell activation, induces dendritic cell apoptosis, drives macrophage polarization to the M2 phenotype, and reduces CD4+/CD8+ T cell and NK cell cytotoxicity	[104,105,106]
Metastasis Promotion	EVs contribute to metastasis by influencing the TME and promoting EMT.	miR-105, miR-6126, miR-7, VEGF, miR-141-3p, miR-205, LIN28	Promotes OC cell proliferation, metastasis, angiogenesis, and EMT	[107,108,109]
Cancer Stem Cell (CSC) Regulation	EVs promote CSC characteristics and facilitate metastasis.	miR-454, Wnt, Notch	Maintains the stemness and self-renewal capacity of CSCs and enhances tumor growth, metastasis, and resistance	[110,111]
DNA Damage Response (DDR) Modulation	EVs transport DNA repair enzymes and regulatory miRNAs.	APE1, POLB, miRNAs	Enhances DNA repair capacity, promotes cancer cell survival, and contributes to tumor progression and therapy resistance	[112]
CAF-Derived EVs and Tumor Behavior	CAF-derived EVs carry pro-tumorigenic molecules.	TGF-β, IL-6, miR-21	Promotes cancer cell proliferation, invasion, metastasis, and chemoresistance	[113,114]
EV Influence on TME	EVs transform stromal cells into CAFs and influence their behavior.	H19, LOXL2, COL1A1, miR-29c-3p	Supports CAF proliferation, activation, and migration and enhances tumor growth and migratory abilities	[115]

**Table 3 biomedicines-12-01806-t003:** The overall mechanisms related to chemoresistance in OC.

Mechanism	Description	Key Players	Impact on Chemoresistance	References
Drug Efflux	Efflux transporters pump cisplatin out of cancer cells, reducing its cytotoxicity.	MDR1 (P-gp), MRP2	Reduced intracellular drug concentration and efficacy	[23,24,27]
Altered Drug Uptake	Changes in transporters limit cisplatin entry into cells.	CTR1	Decreased drug uptake, reduced efficacy	[25]
Epigenetic Changes	Silencing of OTUB2 leads to metabolic reprogramming, enhancing glycolysis and chemoresistance.	OTUB2, SNX29P2, HIF-1α, CA9	Increased tumor progression and chemoresistance	[26]
CAF Involvement	CAFs remodel ECM, facilitate immune cell infiltration, and activate various signaling pathways.	TGF-β, Hedgehog, Notch, Wnt, IL-6, IL-8	Tumor growth, survival, chemoresistance, and immune modulation	[29,30,31,32,38,39,40,41,42,43,44,45,46,47,48]
Extracellular Vesicles (EVs)	EVs from cancer cells and CAFs transfer drug resistance factors and alter recipient cell behavior.	miRNAs, proteins, lipids, mRNAs	Transfer of resistance traits, modulation of the tumor microenvironment, and enhanced survival	[49,52,53]
Apoptosis Pathway Disruption	Altered expression of apoptotic proteins affects cell death.	Bcl-2, Bcl-xL, Mcl-1, Bax, Bak, caspases, IAPs, FLIP	Impaired apoptosis and increased survival	[50,51,52,53,54,55,56,57,58,59,60,61,62]
DNA Repair Pathways	Enhanced DNA repair negates effects of DNA-damaging agents.	HR (BRCA1, BRCA2, RAD51), NHEJ (Ku70/80, DNA-PKcs), NER (ERCC1, XPA, XPB, XPF), BER (OGG1, MUTYH, APE1, XRCC1)	Increased DNA repair, reduced drug efficacy	[63,64,65,66,67,68,69,70,71,72,73,74,75]
Immune Evasion	Cancer cells evade immune detection and destruction.	PD-L1, CTLA-4, TGF-β, IL-10	Decreased immune response and increased tumor survival	[83,84,85,86]
Metabolic Reprogramming	Cancer cells alter metabolism to support growth and survival.	HIF-1α, GLUT1, LDHA, PDK1	Increased glycolysis, lactate production, and resistance to metabolic stress	[87,88,89,90,91]
Tumor Microenvironment	TME influences cancer cell behavior and drug response.	CAFs, immune cells, ECM components, cytokines	Modulation of the drug response, support of tumor growth, and immune suppression	[92,93,94,95,96]

**Table 4 biomedicines-12-01806-t004:** EVs as drug carriers in OC therapy.

Drug/Natural Compound	EV Source	Benefits in OC Therapy	References
Curcumin	Exosomes	Enhanced anticancer effects, improved bioavailability	[127]
Paclitaxel	Exosomes	Enhanced tumor targeting, potential to cross the blood–brain barrier	[128]
Cisplatin	Milk-derived exosomes	Avoidance of drug resistance mechanisms	[128]
Doxorubicin	Exosomes	Reduced cardiotoxicity, improved targeting with antibodies or targeting ligands	[129]
Doxorubicin nanoparticles	Orange-derived EVs	Potent antitumor effects in vivo, inhibition of OC proliferation and angiogenesis	[134]
Amla extract (AE)	SKOV3 cell-derived EVs	Increased miR-375 levels, modulation of tumor microenvironment	[135]
Shikonin (SK)	Exosomes	Reduction in M2 macrophages, suppression of exosome production, inhibition of tumor-promoting immune responses	[136]
Curcumin	Exosomes	Restoration of LncRNA MEG3 levels, reduced miR-214 expression, decreased cisplatin resistance	[137]
Anthocyanidins (Anthos)	Milk-derived exosomes	Improved cisplatin effectiveness, decreased drug resistance, synergistic antitumor effects with paclitaxel	[138]
Various bioactive molecules	MSC-derived EVs	Enhanced therapeutic efficacy, minimized systemic toxicity, targeted drug delivery	[139]

## Data Availability

Not Applicable.

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
