# Peer review of "Extracellular Vesicles in Ovarian Cancer: From Chemoresistance Mediators to Therapeutic Vectors"

_biomedicines, 2024, doi:10.3390/biomedicines12081806_

Round 1

Reviewer 1 Report

Comments and Suggestions for Authors

In this review article, the authors summarized recent literature on the studies of testing natural products in ovarian cancer therapy and the aspects of extracellular vesicles in the chemoresistance of ovarian cancer. These topics have been reviewed in some articles. The niche of this article is supposed to be what was written in the title: the synergistic potential of extracellular vesicles and natural compounds; however, no sessions are discussing this topic. Except it will rewritten otherwise this article lacks novelty.

Author Response

Thank you for your feedback. I have attached the rebuttal letter herewith. 

Reviewer 2 Report

Comments and Suggestions for Authors

The introduction reads very well, the authors manage to introduce the core issues of ovarian cancer in about a page or two, and then they slowly move to the issues of drug resistance. 

Here, some of the paragraphs are nothing but a stub. While there is more text on multidrug resistance formation and the P-glycoproteins, also references by a significant number of citations, the "chapter" on CAFs is not worth mentioning (lines 113 - 120)

Either the authors expand a bit on this very interesting topic, and add newer insights concerning, for example, the existence of subtypes of CAFs that co-exst in cancers, with highly differential functions (also in drug resistance and immune response), or they just entirely skip this part. I would recommend expanding it because there is renewed and constantly growing interest in the issue of CAFs. This also relates to extracellular vesicles or exosomes, which undoubtedly represent one of the mechanisms by which CAFs may increase tumor cell resistance and survival in chemotherapy. 

The next 3 chapters, on apoptosis and DNA damage response/DNA repair,  stem cells, and finally, immune response are a bit more elaborate but also each one of these chapters/paragraphs just barely touches the surface of the matter, and they contrinnute close to nothing to the 2 main topics of the paper ( = EVs and natural compounds).  

The main issue is simply that the authors havent quite decided WHAT EXACTLY they want to write about? Is it the EVs? Or maybe the natural compounds? So why not focus either on the "extracellular vesicles" or "exosomes" , OR you focus on the natural compounds. But both... not possible, unless there is a very close functional connection. But - there isnt.

 In a more focused review, e.g., on EVs, you would have room to stress everything that relates to exosomes/EV within the tumor microenvironment. For this, I would also think CAFs would need broader coverage, as they are a very likely source of EVs in ovarian cancer progression and resistance. What makes CAFs into CAFs, and differentiates them from normal fibroblasts, how is that relevant for EVs ? The role of EVs in immune evasion is briefly mentioned, but again, not elaborated more deeply. 

In short, the title of the review includes the term "extracellular vesicles", but the actual introduction on the molecular functions of EVs is only 17 lines long (from line 192 - 209) and then is taken up later again in the chapter on the diagnostic and prognostic value of EVs (starting from line 459 - 527, about 1 page). None of these 2 disconnected stubs represent a very deep journey into the recent literature on the diagnostic use of EVs, and none of them are n an exhaustive summary of the current literature in the field. 

From here, somewhere in the middle, the review suddently flips over to natural compounds; out of the blue. There is no connection made between EVs and drugs/compounds. However, where is the connection between EVs and natural anti-cancer compounds? This isnt described or discussed ANYWHERE, there simply may be NO CORRELATION,  but shouldnt the title be changed somehow? The title promises things that are not kept, and also, things that are not connected.

It is a bit strange why these 2 aspects are thrown together into the same review, why not make 2 reviews, covering both aspects but properly. 

The introduction of the drug part in this article is a summary of pharmacological developments.  Thats not exactly novel, and it could be illustrated by a timeline, maybe? There aren't any figures in this review, which I find odd. Not a single figure, but a number of rather large tables that have a low "entertainment factor" for the reader. I would definitely think of at least 1-2 figures to be added. 

The interesting bits, and the actual centrepiece of the manuscript, start in line 258 with the chapter on natural compounds. All of these compounds are very non-specific and they bind to or affect many pathways at the same time, one of the resons why the pharma industry is not interested in them; they are anything but "targeted" drugs. 

If the authors aim to make a point that any of these "drugs" my be beneficial for the treatment fo OC they should look into the past and present/ongoing clinical studies. But no clinical trials are mentioned anywhere for any of the compounds mentioned. 

Even a brief research into clinical trials databases, like "clinicaltrials.gov" reveals that there are indeed very few trials in ovarian cancer conducted with any of the compounds mentioned in the text. There is one trial for Curcumin (NCT05306002) and and one on Tanshinone (NCT01452477)  plus a few on resveratrol; although these actually are on polycystic ovary syndrome and not on ovarian cancers. Furthermore, even in the clinical trials description, these are mentioned as "nutritional supplements" and are NOT meant as therapies, or certainly, not primary therapies.  

In my opinion, it should be clearly stated that most or all of the mentioned "natural compounds" are only nutritional supplements, not therapies and that they lack clear molecular mechanisms of action (or they have too many MOIs at the same time), or that they are only antioxidants with a broad spectrum of generalized effects. The title of the review mentions none of these true relationships. 

Furthermore, I dont think this review can be salvaged even by "major revision", in my opinion it should be split into 2 manuscripts and each one of the should be handled with more focus, and in more depths. 

Comments on the Quality of English Language

there are no big issues with the english language use; this isnt problematic.

Author Response

(The authors gave the same response as above.)

Round 2

Reviewer 1 Report

Comments and Suggestions for Authors

The authors have addressed the previous concerns.

Author Response

Dear Reviewer 1,

Greetings!

We are pleased to submit our revised manuscript titled “Extracellular Vesicles and Natural Compounds in Ovarian Cancer: Insights and Intersections” for publication in your esteemed journal. We have highlighted the changes that was made in the manuscript according to the recommendation from the reviewer using point-by-point format.

The authors have addressed the previous concerns.

Reply: Thank you for your feedback. We appreciate your comments and are glad to hear that the authors have addressed the previous concerns. Your input has been invaluable in improving the quality of the manuscript.

Thank you for the opportunity to further explain our research hence we request you to consider the manuscript for publication in the esteemed journal and oblige.

Thank you.

Sincerely,

Barathan Muttiah

Reviewer 2 Report

Comments and Suggestions for Authors

The authors have tried very hard to save this review from rejection. I have been rather strict in the last review and I still think the review was not even close to deserving acceptance in its previous version. 

I am still not convinced that the "natural compounds" mentioned in this review, which are all nutritional additives, will even play a  role in clinical chemotherapy; simply for the reason that there are no trials showing any benefits. The question WHY there are no triels is beyond the scope of this review, and the manuscript, it is more related to the fact that "alternative therapies" are not gaining the attention of most clinicians nor the pharmaceutical industry, for obvious reasons.

Nevertheless, the authos have added significant information and several pages of new material trying to establish a link between the basically unrelated topics of the paper: extracellular vesicles, and "natural compounds". 

These links are substantiated by a considerable number of references illustrating the benefits of EVs as vehicles for drug delivery. 

The authors have added an almost 2 pages paragraph on the role of CAFs in ovarian cancer, and this is reasonably supported by references. This is in response to my own request to deal with the role of CAFs more seriously. The authors also added about 1 page on apoptosis and 1 page on DNA repair mechanisms. All of this is nice and improves the review (probably). It still doesnt help to improve the FOCUS of the review. 

Starting from line 387, the authors then finally further elaborate on one of the 2 min topics of this review: the extracellular vesicles (EV). This is a critical paragraph that should represent the core of the manuscript: it should also be a synthesis of the topics mentioned before (apoptosis, CAFs, DNA damage response, stem cells) and the EV problematic. 

Therefore, this section deserves to be a chapter on its own, and its central importance must be more stressed and clarified: this is the syntheis of the previous paragraphs and justifies why we are talking about EVs here. This goal can still be improved by actually referring more to the topics outlined before. This shouldn't be too difficult to stress more obviously: simply by pointing out WHERE EXACTLY it likes to the topics of apoptosis, CAFs, DNA damage response, and stem cells in ovarian cancer. Currently, there is a strong focus (in this paragraph) on non-coding RNAs, migration, invasion, immune checkpoint evasion, cytokines and chemokines, and cell proliferation, but not on apoptosis, CAFs, DNA damage response, and stem cells. With other words, the narrative within the manuscript is inconsistent and jumps from one topic to another, this still has to be improved. 

Figures 1 and 2 are very nice and easily understandable additions to the manuscript. 

Finally, the authors have added a massive section on the use of EVs as vehicles for drug delivery, starting from page 732. The "trick" to talk about the use of EVs for delivery on natural compounds, not synthetic and highly potent chemotherapeutic drgs like taxanes, appears a little bit artificial. One of the main strategies why nanoparticles (NPs) or EVs are used for drug delivery is the idea to specifically target cancer cells and spare non-cancerous, normal cells in the body, thereby circumventing damaging healthy tissues in therapy. This is NOT necessary when natural compounds are used that don't have pronounced specificity for cancer-relevant pathways (or multiple targets/pathways at the same time), so the rationale to use EV for delivering things like curcumin appears somewhat counterproductive. Indeed, using NPs and/or EVs do deliver highly cytostatic drugs such as doxorubicin, paclitaxel, and cisplatin are mentioned and supported by references. Those are fully logical approaches in OC. 

But why would EVs/NPs be used to transport and deliver natural compounds like anthocyanidins? I think there is very little evisdence that EVs/NPs have been used for such purposes. And indeed, there are no references given for this specificity.  Instead, what is described is the impact of certain natural compounds, like AMLA extract, on the generation and secretion/release of EVs in the tumour microenvironment. Meaning, the actual reverse of whats in the title and also the title of the chapter. This, the actual use of EVs for delivering natural compounds is stull missing, or very thin indeed. This needs to be strengthened and improved, if there is such existing evidence. 

Comments on the Quality of English Language

there are no major issues with english language 

Round 3

Reviewer 2 Report

Comments and Suggestions for Authors

The authors have taken my recommendations to provide more focus to the manuscript very seriously. <they have indeed reorganized the manuscript significantly and focused now a lot more on the topic of extracellular vesicles in ovarian cancer. This was my major concern, and it is now very significantly improved. <z>At the same time, the authors have more or less sacrificed the chapter on "natural compounds" which - first of all - were mostly food additives and no drugs, and they havent contributed much to the novelty and the informative content of this manuscript. I would consider this as "good riddance". 

<There are now mostly "small issues" that need to be fixed, but which would contribute to the quality of the manuscript. 

lines 150 - 153: While iCAFs, myCAFs, apCAFs, and even vCAFS (vascular CAFs) are quite widely accepted, i am  not so sure about CAF "subtypes" mentioned in the manuscript. These would need a more substantial description and maybe more references to validate their existence in OC. This includes the following: developmental CAFs, CD146+ CAFs, α-SMA+ CAFs, asporin+ CAFs, and versican+ CAFs. What is the evidence of these CAF subtypes beyond the one manuscript mentioning them (Ref. 32), how are they overlapping o identifcal with the more widely established CAF subtypes? 

Table 1 is useful, but its rather generic and general information, so the information content is limited. 

In contrast, I do not find table 2 useful or informative at all. Esppecially, applications mentioned (like Delivery of natural compou-nds (berry anthocyanidins)) arent useful as long as there are no references in the table. And almost all of thse elements are nothing more than keyword, without much substance. I would either refine and expand table 2 to make it useful - or just skip it. 

The chapters 3 and 4 on EVs arent properly defined, and the headlinesare too similar to make sense: 

Chapter 3. Extracellular vesicles as emerging drug delivery 

Chapter 4. Challenges of EV as drug delivery agent

Both chapters deal with drug DELIVERY. Whats the difference? What is the specificity of each chapter? Why should I read it when they sound the same. 

The loading of EVs with any drugs seems not ery advanced, at the moment (Im not an expert); how do researchers aim to put specific cargo into EVs? HAre there robust and reproducible methods for this already? Itf these exist, they need to be explained in more details in chapter 4. 

The loading of EVs for the purpose of drug delivery is also the subject of chapter 3 (lines 549 to 575). 

As this is the central and probably most important element of the review, this paert still needs more substance to be truly up to date and informative for researchers interested in this topic. At the moment, it still scratches the surface. 

Comments on the Quality of English Language

English language quality has not been a big issue in this manuscript and theres nothing that cannot be fixed after acceptance of the manuscript. 
